# The Influence of Trait Emotional Intelligence on Archers’ Autonomic Cardiac Recovery Responses Immediately After a Shooting Session

**DOI:** 10.3390/bs9050055

**Published:** 2019-05-20

**Authors:** Nihal Dal

**Affiliations:** Faculty of Sport Sciences, Manisa Celal Bayar University, Halil Erdoğan St., Manisa 45040, Turkey; nihal_arc@yahoo.com or nihal.dal@cbu.edu.tr

**Keywords:** archery, archers, heart rate variability, trait emotional intelligence

## Abstract

The present study aimed to investigate the association between trait emotional intelligence and heart rate variability (HRV) recovery responses of archers immediately after a shooting session. The sample included 87 novice archers ranging in age from 18 to 26. Participants first completed Schutte Emotional Intelligence Inventory. Then, they shot 10 arrows from 18 m to an 80-cm diameter target in four minutes. Afterward, participants’ HRV recovery responses were measured during a four-minute recovery period. In this study, HRV was represented in terms of low frequency (LF), high frequency (HF), and LF/HF ratio. Results indicated a significant relationship between emotional intelligence and HRV recovery responses. A regression model containing emotional intelligence sub-dimensions was able to explain a significant amount of variance in HRV frequency domain parameters. Besides, high emotional intelligence archers were found to have higher-level LF and HF power but a lower LF/HF ratio than their low emotional intelligence counterparts. Taken together, the results observed in the present study indicated that emotional intelligence might give rise to more adaptive HRV recovery responses following a demanding arrow shooting session.

## 1. Introduction

Understanding the link between athletes’ emotional experience and physiological responses is one of the most central aims of sport psychology. For this purpose, several theoretical frameworks have been offered, including Multi-Dimensional Anxiety Theory [1] and Catastrophe Theory [2]. Generally speaking, these theories focused primarily on athletes’ pre-competition anxiety as well as bodily responses and their impact on performance. Sport psychology literature also emphasized the importance of other emotional experiences rather than anxiety and regulation of these emotions for better performance. In this respect, as the emotional skills that can facilitate athletic performance such as establishing and maintaining appropriate emotional conditions and managing emotions [3,4] overlap with emotional intelligence, this relatively recent psychological construct may be useful in understanding athletes’ emotional and bodily responses. Before presenting a literature review regarding emotional intelligence in sports, readers should be aware of the theoretical distinction between trait and ability emotional intelligence proposed by Petrides and Furnham [5,6]. In this respect, “trait EI (or emotional self-efficacy) refers to a constellation of behavioral dispositions and self-perceptions concerning one’s ability to recognize, process, and utilize emotion-laden information, and Ability EI (or cognitive–emotional ability) refers to one’s actual ability to recognize, process, and utilize emotion-laden information” [7] (p. 278). Considering the previous statement, it seems that despite their similar theoretical basis, trait and ability emotional intelligence are different constructs. One of the most prominent distinctions between trait and ability-based emotional intelligence stems mainly from the measurement methods. While ability-based emotional intelligence is measured via performance tests, trait emotional intelligence is measured through self-report tests. Although Mayer et al. argued that self-report ability and actual ability are minimally correlated, the present study focuses on trait emotional intelligence measured via self-report tests. In this way, it might be possible to provide some evidence on whether trait emotional intelligence can be a useful construct for researchers and practitioners in the field of sport psychology. 

Previous research showed that both trait- [8] and the ability- [9] based measures of emotional intelligence might have the ability to predict athletic performance. Moreover, research findings also indicated that trait emotional intelligence might predict athletes’ physiological responses to stress [10,11]. Despite the wealth amount of study documenting the effect of emotional intelligence on athletes’ emotional and bodily responses before and during a competition, researchers simply ignored athletes’ post-competition biological responses and their association with emotional intelligence. Post-competition physiological responses of an athlete are of both theoretical and practical importance due to their role in athletes’ life quality as well as recovery for new performance. Psychophysiological recovery immediately following the performance is of vital importance in archery, as the archers have to shoot 72 arrows in 12 shooting series. Therefore, factors with a potential to influence archers’ psychophysiological recovery deserve a careful examination in well-controlled experiments. The present study aimed to explore whether emotional intelligence may account for the athletes’ psychophysiological recovery, as represented by heart rate variability (HRV). HRV is an essential marker of autonomic nervous system (ANS) activity modulation [12]. The ANS is modulated by two different well-balanced systems: the sympathetic nervous system (SNS), which is related to the fight or flight response; and the parasympathetic nervous system (PNS), which is associated with rest and the digestive system [13]. Spectral analysis of HRV is one of the essential methods for understanding the influence of SNS and PNS in the regulation of the cardiovascular system. According to Massimo Pagani’s HRV model [14,15,16], three components represent SNS, PNS, and the balance between SNS and PNS. Thus, high frequency (HF) power (0.15 to 0.40 Hz) represents cardiac parasympathetic tone. On the other hand, low frequency (LF) power (0.04 to 0.15) represents cardiac sympathetic outflow. The sympathovagal tone is determined by the LF/HF ratio [17]. 

Previous research demonstrated that HRV might be a marker for the attentional workload during athletic performance [18]. Further, research findings revealed a link between HRV and athletes’ psychological states under pressure [11]. Taken together, it seems that HRV might have the potential to reflect athletes’ psychophysiological recovery immediately after a performance. 

The present study aimed to examine whether trait emotional intelligence may predict archers’ psychophysiological recovery immediately after a shooting session. As the archery is a both physically and psychologically demanding activity, psychological as well as physiological recovery play a vital role in preparation for the next shooting session. Moreover, emotional intelligence skills such as understanding and managing one’s own emotions [19] can have an important effect on archers’ shooting precision. To the best of my knowledge, no previous study examined whether certain psychological qualities, trait emotional intelligence in this case, might be associated with archers’ HRV recovery responses. As psychophysiological recovery is highly important in the 12 shooting sessions, such information may be beneficial for athletes and coaches. In light of the research mentioned above, I hypothesized that emotional intelligence including psychological skills such as managing one’s own emotions should predict archers’ psychophysiological recovery as represented by HRV. 

## 2. Materials and Methods

### 2.1. Participants 

Participants were 86 college students ranging in age from 18 to 25 years (mean = 3.13, (SD = 1.34)). None of the participants had previous experience in archery. However, they enrolled in an archery class to receive course credit. At the end of the 14-week course, individuals having a final exam grade higher than 70 were invited to take part in the study. The author personally invited all participants to take part in the study. Individuals who agreed to participate in the study gave informed consent. The local ethics committee approved the study protocol (Manisa Celal Bayar University, Institute of Health Sciences Ethics Committee, Ethics number: 20.478.486).

### 2.2. Measures

#### 2.2.1. Emotional Intelligence Scale

The Schutte Emotional Intelligence Scale, developed by Schutte et al. [20], revised by Austin et al. [21], and adapted for the Turkish population by Tatar et al. [22], was used to measure emotional intelligence. The scale contains 41 items and generates an overall EI score, as well as scores for three subscales: regulation of emotion, appraisal of emotion, and utilization of emotion. Regulation of emotion measures the extent to which people report being able to control their own and others’ emotions; utilization of emotion measures the extent to which people report being able to use emotions in solving problems; and appraisal of emotions measures the extent to which people report being able to identify their own and others’ emotions. The internal consistency score for the present sample was 0.87. 

#### 2.2.2. Heart Rate Variability (HRV) Measurement and Analyses

A Nexus-10 system and its supplied software (Biotrace+, Mind Media CV, the Netherlands), was used to measure HRV derived from electrocardiogram (ECG) in a lead II configuration.

The ECG signals were saved at 16-bit resolution with a sampling rate at 1024 Hz. The HRV of each participant was obtained from the time series of beat-to-beat intervals, RR intervals, that was immediately extracted from the ECG data. Before HRV analyses, the RR intervals were visually inspected, and artifacts were corrected. Then, the corrected RR intervals were converted to interbeat intervals (IBI) time series and subjected to fast Fourier transform (FFT) to compute frequency domain indices of HRV.

#### 2.2.3. Procedure

Participants first completed the Schutte Emotional Intelligence Inventory. Then, participants shot 10 arrows in 4 minutes from 18 m to an 80-cm-diameter target. The shooting task was self-paced, so participants decided when to shoot an arrow and how long to prepare to shoot. Based on a pilot study of five novice archers, we determined that participants were required to shoot an arrow in approximately 20–25 seconds. Participants were given additional arrows if they finished their shooting in less than 4 minutes. At the end of the shooting session participants’ HRVs were measured for 4 minutes. All experiments were conducted in Manisa Celal Bayar University, Faculty of Sport Sciences Archery Facilities. Only one participant and the experimenter was present during the experiments. The experimenter provided all shooting and safety equipment to the participants. 

#### 2.2.4. Statistical Analysis

To explore the relationship between emotional intelligence sub-dimensions and HRV parameters, Pearson correlation coefficients were calculated. Afterward, regression analysis was performed to test whether the regression model containing emotional intelligence sub-dimensions may have the ability to predict archers’ HRV immediately after a shooting session. Lastly, the sample was grouped into high and low emotional regulation groups based on a median split of the emotional regulation sub-dimension score in the Schutte Emotional Intelligence Inventory. Then, a one-way multiple analysis of variance was carried out to compare the HRV frequency domain of LF, HF, and LF/HF ratio. 

## 3. Results

### 3.1. The Relationship between HRV and Trait Emotional Intelligence

As illustrated in Table 1, results of the Pearson correlation analysis revealed that among the emotional intelligence sub-dimensions only the regulation of emotions was positively and significantly correlated to LF ( r = 0.30, *p* = 0.005) and HF (r = 0.33, *p* = 0.002). On the other hand, there was a negative and significant correlation between the regulation of emotions and the LF/HF ratio (r = −0.29, *p* = 0.007). 

### 3.2. The Ability of Trait Emotional Intelligence to Predict HRV

In the next stage of the experiment, three regression analysis with the enter method tested whether the emotional intelligence sub-dimensions may have the ability to explain the variation in archers’ HRV during the recovery period. As illustrated in Table 2, Table 3 and Table 4, results showed that the emotional intelligence sub-dimensions could explain a significant amount of variance in LF, HF, and LF/HF ratio. The regulation of emotions sub-dimensions of emotional intelligence was the strongest predictor of the HRV. 

### 3.3. HRV Differences between High and Low Emotional Regulation Individuals. 

Lastly, based on the observed power of the emotional regulation sub-dimension in predicting HRV, I decided to examine whether HRV may differ as a result of emotional regulation. For this purpose, participants were grouped into either high or low emotional regulation groups based on a median split of the regulation of emotions sub-dimensions’ scores of all participants. Then, I performed a multiple analysis of variance (MANOVA) to compare HRV parameters of LF, HF, and the LF/HF ratio between high and low emotional regulation groups. According to Pillai’s trace, the multivariate analysis of variance revealed a significant main effect for the regulation of emotions (V = 0.156, F(3, 82) = 5.04, *p* = 0.003, η2 = 0.16). Separate univariate analyses of variance of the dependent variables revealed that high emotional regulation individuals have higher LF (F(1, 84) = 6.38, *p* = 0.013, η2 = 0.07) and HF (F(1, 84) = 9.73, *p* = 0.002, η2 = 0.10) than their low emotional regulation counterparts during the recovery period immediately after arrow shooting. However, low emotional regulation individuals had higher LF/HF (F(1, 84) = 9.93, *p* = 0.002, η2 = 0.11) ratio than high emotional regulation individuals.

## 4. Discussion

The present study aimed to explore whether emotional intelligence may have an account for the archers’ HRV activity immediately after a shooting session. Results of the study provided preliminary evidence that emotional intelligence, specifically its emotional regulation dimension, may have the ability to predict archers’ autonomic cardiac activity represented by HRV following an arrow shooting session which requires a considerable amount of physical and mental effort. In this respect, results revealed that archers high in emotional regulation skills have greater HRV than their low emotional regulation counterparts. The observed results suggested that emotional intelligence may give rise to more appropriate recovery responses during a recovery period.

The first finding that I want to deal with is the greater level of LF power observed in the high emotional regulation group. LF power reflects sympathetic outflow [14]. The sympathetic nervous system increases heart rate, constricts blood vessels, and decreases gastrointestinal motility [23]. At onset and during steady state exercise the sympathetic nervous system predominates over the parasympathetic nervous system [24]. Accordingly, it seems that emotional intelligence may give rise to more appropriate cardiac responses immediately following a physically and psychologically demanding task, arrow shooting in this case.

Moreover, a higher level of LF power may be indicative of archers’ effort to control their respiration, which is vital for more accurate shooting. Hence, a higher level of LF power which reflects sympathetic activity suggested that archers high on emotional regulation may more effectively control their respiration. In other words, high emotional intelligence archers executed the skills more effectively, with better arrow shooting. In a previous study, similar results were observed indicating a higher level of LF power in experienced archers [25]. In the light of the observed results together with previous results by Carrillo et al. [25], it seems that heightened sympathetic activity may be observed in more experienced and emotionally intelligent archers. 

However, our results are not in line with some other previous results suggesting that performance in archery is inversely associated with the sympathetic activity [26]. Despite this finding, the present study and another previous study provided preliminary evidence that archers with a higher level of experience [25] and emotional regulation skill may have a higher sympathetic activity which leads to an increased level of arousal. Actually, in demanding physical activities such as archery, it is almost impossible to prevent an increase in sympathetic activity. Therefore, researchers should consider examining factors that can affect the compensation of a higher level of arousal. Results of the present study indicated that specific emotional skills might facilitate a higher level of sympathetic activity or arousal during recovery.

Though previous studies examined trait emotional intelligence mainly concerning pre-competition emotional responses [27,28], the results observed in the present study indicated that trait emotional intelligence might also be associated with post-competition responses. Most of the sport-specific psychological theoretical frameworks mentioned earlier focused primarily on athletes pre-competition responses. However, the observed results of the present study provided preliminary evidence for the necessity of the post-competition responses. Taken together, it seems that both researchers and practitioners might consider trait emotional intelligence as a likely factor to understand athletes’ responses immediately after the performance.

## 5. Conclusions 

Findings of the present study may have several implications for researchers as well as practitioners such as coaches and athletes. First, coaches or movement scientists should be aware of psychological factors, emotional intelligence in this case, that can influence athletes’ physiological responses. Further, researchers should consider examining whether specific individual psychological differences may give rise to better tolerance of physiological arousal. Lastly, the present results provided preliminary evidence that trait emotional intelligence may account for the athletes psychophysiological responses immediately after the performance. 

## 6. Limitations 

Although the present study provided some preliminary evidence for the effect of emotional regulation on archers physiological recovery, it also includes several limitations. First, this study included only novice archers, limiting the generability of the results. Second, the respirational activity which is vital for the HRV was not measured in the present study. In future studies, researchers may assess respiration in addition to HRV for a better understanding of the topic. 

## Figures and Tables

**Table 1 behavsci-09-00055-t001:** The relationship among trait emotional intelligence and heart rate variability (HRV). LF: low frequency; HF: high frequency.

	LF	HF	LF/HF Ratio
Regulation of Emotions	0.300 *	0.321 **	0.290 *
Appraisal of Emotions	0.002	−0.075	0.002
Utilization of Emotions	0.020	0.011	−0.147

* *p*<0.01; ** *p*<0.05.

**Table 2 behavsci-09-00055-t002:** The ability of the trait emotional intelligence to predict LF power during recovery.

	B	β	t	R	R^2^ _adj_
Constant	−3690.34		−0.834	0.335	0.11
Regulation of Emotions	304.34	381	3.21 *		
Appraisal of Emotions	−141.41	−0.096	−0.789		
Utilization of Emotions	−91.61	−0.100	−0.806		

* *p* < 0.05.

**Table 3 behavsci-09-00055-t003:** The ability of the trait emotional intelligence to predict HF power during recovery.

	B	β	t	R	R^2^ _adj_
Constant	−4902.79		−1.04	0.400	0.13
Regulation of Emotions	386.18	444	3.85 *		
Appraisal of Emotions	−326.001	−0.203	−1.72		
Utilization of Emotions	−84.90	−0.100	−0.713		

* *p* < 0.05.

**Table 4 behavsci-09-00055-t004:** The ability of the trait emotional intelligence to predict LF/HF power during recovery.

	B	β	t	R	R^2^ _adj_
Constant	8.040		3.34	0.324	0.07
Regulation of Emotions	−0.136	−0.315	−2.64 *		
Appraisal of Emotions	130	0.164	1.34		
Utilization of Emotions	−0.043	−0.088	−0.702		

* *p* < 0.05.

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
