# Peer review of "The Influence of Trait Emotional Intelligence on Archers’ Autonomic Cardiac Recovery Responses Immediately After a Shooting Session"

_behavsci, 2019, doi:10.3390/bs9050055_

Reviewer 1 Report

Thank you for the opportunity to review this manuscript.

Lines 30-32: Need citations.

The Introduction need to be extending to cover better the topic, especial about trait emotional intelligence related to specificity of archery sport.

Participant: Why you choose to investigate only students without experience in archery? Why you do not compare them with one control group of students with sport archery background. What were the inclusion and exclusion criteria’s?

The Discussion need to be improved to sustain your findings related to the findings of previous studies.

I recommend highlighting the strength and limits of your study. 

The conclusions need to be revised to underline the main ideas of your study. In present form the conclusions are too generally and do not highlight the main evidences of study.

Author Response

I want to thank you for your valuable contribution to the paper. Below you may find the responses to your suggestions.

Yours Sincerely

Responses to reviewer 1

The Introduction need to be extending to cover better the topic, especial about trait emotional intelligence related to specificity of archery sport.

I tried to extend the introduction section based on your suggestion. I tried to make a distinction between trait and ability conception of emotional intelligence. Further, I tried to clarify why trait emotional intelligence may be associated with archers’ psychophysiological responses.

Participant: Why you choose to investigate only students without experience in archery? Why you do not compare them with one control group of students with sport archery background. What were the inclusion and exclusion criteria’s?

Archery is a costly and risky athletic discipline. Hence, it was not easy to find enough amount of experienced archers to conduct an experiment. Further, I believe that participants in this experiment had almost the same level of experience (14 weeks, 2 hours per week) which makes them a more homogeneous cohort. Moreover, the examination of the beginner archers’ responses might provide valuable information for practitioners.

The Discussion need to be improved to sustain your findings related to the findings of previous studies.

I tried to improve the discussion section following your suggestions.

I recommend highlighting the strength and limits of your study. 

I added a limitation section to the paper.

The conclusions need to be revised to underline the main ideas of your study. In present form the conclusions are too generally and do not highlight the main evidence of study.

I tried to revise the conclusions following your suggestions.  

Reviewer 2 Report

Dear Author,

Thank you for the opportunity to review this manuscript. In the attached file you will find the review form.

Sincerely,

Reviewer

Author Response

I want to thank you for your valuable contribution to the paper. Below you may find the responses to your suggestions.

Yours Sincerely

Responses to reviewer 2

The introduction section needs to be unpacked. More scientific studies in relation to the analyzed problem should be provided and discussed regarding previously published articles. Moreover, explain more about emotional intelligence and specifics of it. What is the definition of emotional intelligence and how it relates to your problem? What are the emotional skills? It is a broad definition. Please clarify.

Based on your suggestions I tried to extend the literatĂĽre review, especially regarding emotional intelligence. I tried to emphasize the importance of the study. The last paragraph of the introduction section describes the hypothesis and purpose of the study.

There are numerous of similar studies, why particularly your study is important?

Despite the second reviewer's statement which is shown below, to the best of my knowledge, no previous study tested post-performance psychophysiological recovery responses in relation to emotional intelligence, especially in beginner archers. However, I tried to emphasize why his study is of importance.   

Participants. The section on participants lacked some critical information that could have informed the reader about the study sample. The demographics of the participants and how they are recruited were not reported. We know that the age was 18-25 years, but the mean age of participants was unknown.

Means and standard deviations were included for the participants' age. Further, I explained how the participants were recruited.

What specific method of participant selection you have used?

How participants were selected was described detailly in the method section.

The process of the study. Please explain the procedure in more depth. How they were researched? The place, the time. Explain participants agreed to participate in the study process. What specific information the particippant content included?

The ethical aspects of the study? Please describe what specific steps you have included in ensuring the safety for research participants. The local ethics committee approval was declared, but the overall process was not mentioned.

I described the experiment procedure and safety regulations. Experiments were conducted in a facility that specifically designed for the archery. Participants were tested one by one, so, there was no opportunity for an accident. Participants also used standard self-protection equipment. Moreover, the experimenter (the author) was a former top-class international archer who is an expert on this specific athletic discipline.

The results of your study in the discussion part need to be unpacked and discussed in more depth in comparing with other studies in a similar field.

2. The discussion is thin in content as it almost simply repeats what was reported in the Results section with some citations of literature which are mostly not fully relevant to the discussion.

3. The discussion part should e discussed and linked with the theoretical framework, the theories that were discussed in the introduction section.

I tried to do my best in revising the discussion section. However, the theoretical frameworks mentioned in the introduction section does not provide theoretical support to my results, as they focused on pre-competition anxiety or emotional responses. I emphasized in the discussion section that in addition to pre-competition responses, post-competition responses can be predicted via trait emotional intelligence.  

Best Regards

Round  2

Reviewer 1 Report

The authors improved the manuscript.